# Ethics of Smart Cities: Towards Value-Sensitive Design and Co-Evolving City Life

Dirk Helbing [1,2,*,†], Farzam Fanitabasi [3], Fosca Giannotti [4], Regula Hänggli [5], Carina I. Hausladen [1], Jeroen van den Hoven [6], Sachit Mahajan [1], Dino Pedreschi [7] and Evangelos Pournaras [8]

1 Computational Social Science, Department of Humanities, Social and Political Sciences, ETH Zürich, 8092 Zürich, Switzerland; carina.hausladen@gess.ethz.ch (C.I.H.); sachit.mahajan@gess.ethz.ch (S.M.)
2 Complexity Science Hub Vienna, 1080 Wien, Austria
3 Department of Communication Science, Faculty of Social Sciences, Vrije Universiteit Amsterdam, 1081 HV Amsterdam, The Netherlands; f.fanitabasi@vu.nl
4 ISTI-CNR, 56124 Pisa, Italy; fosca.giannotti@isti.cnr.it
5 Department of Communication and Media Research, University of Fribourg, 1700 Fribourg, Switzerland; regula.haenggli@unifr.ch
6 Faculty of Technology, Policy and Management, TU Delft, 2628 BX Delft, The Netherlands; M.J.vandenHoven@tudelft.nl
7 KDD Lab, University of Pisa, 56126 Pisa, Italy; dino.pedreschi@unipi.it
8 School of Computing, University of Leeds, Leeds LS2 9JT, UK; E.Pournaras@leeds.ac.uk
* Correspondence: dirk.helbing@gess.ethz.ch
† Current address: Stampfenbachstrasse 48, 8092 Zürich, Switzerland.

**Abstract:** The digital revolution has brought about many societal changes such as the creation of "smart cities". The smart city concept has changed the urban ecosystem by embedding digital technologies in the city fabric to enhance the quality of life of its inhabitants. However, it has also led to some pressing issues and challenges related to data, privacy, ethics inclusion, and fairness. While the initial concept of smart cities was largely technology- and data-driven, focused on the automation of traffic, logistics and processes, this concept is currently being replaced by technology-enabled, human-centred solutions. However, this is not the end of the development, as there is now a big trend towards "design for values". In this paper, we point out how a value-sensitive design approach could promote a more sustainable pathway of cities that better serves people and nature. Such "value-sensitive design" will have to take ethics, law and culture on board. We discuss how organising the digital world in a participatory way, as well as leveraging the concepts of self-organisation, self-regulation, and self-control, would foster synergy effects and thereby help to leverage a sustainable technological revolution on a global scale. Furthermore, a "democracy by design" approach could also promote resilience.

**Keywords:** smart cities; digital democracy; participation; co-creation; sustainability; resilience

## 1. Introduction

In recent years, the idea of "smart cities" has increasingly been adopted around the world. A "smart city" can be defined as a city, in which information and communication technology (ICT) is integrated with the physical, social, and business infrastructure for the sake of optimisation, automation, efficiency and control [1] However, a number of contrasting views on smart cities initiatives have emerged over the years. Some researchers consider it to be the solution to the challenges related to urbanisation, sustainable development, as well as social and economical well being [2]. Others claim the idea of smart cities is ambiguous and dystopian [3]. Despite the differing views, smart city projects have spread across the globe. Current smart city projects usually exploit big data and the Internet of Things (IoT) to create a more efficient urban environment. When using these technologies primarily for automation and control, they may threaten freedom and

privacy of its citizens [4–6]. However, these technologies can also be used to empower citizens, unleash collective intelligence, and create value [7] by promoting principles of open technology and social innovation [8–10].

Therefore, one aim of this paper is to stress that planning, optimisation and control are not the only possible uses of digital technologies, and probably not the best ones, when it comes to managing complex dynamical systems [11]. Instead, and this is another aim of this paper, we propose to organise the digital world, and smart cities in particular, in a distributed and participatory way, which can support resilience and other desirable features. Furthermore, this paper explores how promoting human, ethical, cultural and democratic values can foster a pathway towards greater sustainability. Technology should reflect a "citizens first" attitude through open technology, digital rights, and an inclusive and collaborative environment.

This paper covers the literature on complexity science, ethics of information technology, interactive and sustainable smart cities, and research on deliberative democracy. Compared to previous publications, the novelty of this paper lies in connecting these four strands: We interpret cities and societies as living entities and, therefore, self-adaptive and self-organising systems (Section 2). This helps us to provide recommendations on how to design fair and responsible AI for social good (Section 3). We also provide a uniquely compiled set of examples of what digital sovereignty could look like in an urban setting (Section 4).

The following parts of this paper are structured as follows: The first part, mainly Section 2, describes the development of information technologies shaping the concept of smart cities. The second part, comprised of Sections 3–5, point out how designing technologies for values could promote a more sustainable pathway of cities that better serves people and nature. Section 6 offers a summary, conclusion, discussion, and outlook.

## 2. Early Use of Information Technologies in Cities

Since the very beginning of their existence, computers have been used to simulate and improve cities and mobility. Traffic simulation models, for example, were around already in the 1950's [12]. Traffic light control, as well, applied computers early on [13]. Later on, auto-CAD programs were used for building design [14–17] and urban planning [18–24]. Eventually, digital technologies became pervasive. By now, they are used to monitor, manage and control cities on all imaginable levels and scales. In this connection, there are increasing efforts to build digital twins [25].

### 2.1. Automated Cities

The birth of the idea of "smart cities" was a logical consequence of this development [26]. Some of the earliest attempts were put forward by IBM [27] and Cisco [28]. IBM proposed the Smarter Planet Project [27] and the Smarter City Challenge [29]. In 2008, Cisco detailed the "Connected Urban Development program" and dedicated USD 15 million to it [28]. However, by the year 2020, the company quit its efforts to digitise cities [30].

Before this decision was taken, a data-driven vision of cities had spread, assuming ubiquitous sensing, using the IoT. In this way, everything from traffic flow to logistics and waste disposal would be optimised and automated [31]; people would be surveilled as well. Eventually, in Germany a "Smart City Charter" was drafted, which partly went so far as to suggest a post-decision and post-voting society [32] (p. 43). According to this vision, societies would be run in a data-driven way, using Big Data and powerful Artificial Intelligence (AI).

In fact, so far AI seems to hinder rather than to facilitate a value-sensitive deployment of technology: Most AI models are black box models. Therefore, designers have only limited possibilities to comprehend whether AI systems perform the tasks at hand in ways that embody the relevant values or not [33]. This may partly explain why, despite all efforts and many billions spent, automated society did not meet the expectations, yet. In fact, an evaluation of smart cities projects resulted in a rather disillusioned view [34].

Moreover, in the year 2019, only one Silicon Valley city—San Francisco—had made it into only one (out of five) top ten lists of the world's most liveable cities [35–39] (San Francisco was ranked 9th on the Deutsche Bank Liveability Survey [38]). Furthermore, smart cities projects around the world are still far from being sustainable, even though the political goal is to reach sustainability by 2030 [40], i.e., in less than a decade.

Further problems result from the mass surveillance approach underlying many smart city concepts. According to Germany's "Smart City Charta", "[b]ehavioral data can replace democracy as the social feedback system" [32] (p. 43), as long as enough data are collected. If the collected data were managed by the private sector, this example would illustrate all three normative concerns identified by the ethics-focused smart city literature: privatisation, platformisation and domination [41]. According to our judgement, such concepts for the society of the future are based on privacy intrusion, profiling, targeting, behavioural manipulation, and discrimination. Hence, they are likely to significantly interfere with fundamental human rights.

### 2.2. Digital Twins

By 2030 it is expected that around 50 billion measurement sensors of various kinds will be used worldwide, many of them connected to the Internet [42]. This establishes the IoT [43], which enables smart homes and smart cities.

Given the vast amount of data produced by the IoT, it has become fashionable to produce "digital twins" or "digital doubles" also of human beings. For infrastructures, traffic, production, or climate, such models of reality can be certainly helpful [44–46]. However, there are some risks and challenges. First, it is crucial how the data are used, and who has access to them for what purposes. A war room approach would certainly be inappropriate for the use of these data [47]. Second, the application of digital twin technology to individual lives, personalities, or health is particularly tricky. While

- Cyber threats such as hacking;
- False positives or negatives, wrong classifications;
- Misleading patterns and spurious correlations in the data;
- Overfitting ("not seeing the forest for the trees");
- Problems of calibration, validation, sensitivity, or convergence;
- Wrong interpretations or lack of explanatory understanding;

are not uncommon when applying digital twins to humans or things, the following issues are especially problematic in connection with humans:

- Violations of privacy and human rights;
- The possible misuse of data (e.g., for manipulation);
- Related psychological and health risks;
- Distraction from the relevant details ("attention economy");
- Discrimination;
- Inadequate simplifications;
- Ignorance of relevant aspects (e.g., non-measurable qualities);
- Treatment of subjects like objects;
- Confusing "digital twins" (the picture) with reality;
- "butterfly effects" (according to which small details may matter in complex dynamical systems).

As a consequence of the above, the wrong use of a "digital twins" approach may cause considerable damage: It may undermine the natural adaptiveness and emergence that are characteristic for living systems, also cities. We, therefore, want to caution against overwriting the functional principles of self-organising systems by a new "operating system", using AI-based control. This could produce serious malfunctions or major damage. Instead, a complexity science approach is needed, which considers cities—to some extent— as self-organising, self-regulating and co-evolving systems [25]. It must also be considered

that a merely data-driven approach may eliminate desirable qualities, which may affect anything from human dignity to culture to quality of life.

Given the above, one pressing question to answer is how to manage and use the massive amounts of data now produced by the Internet of Things about people and environment. Should data be centrally stored by governments or businesses and allowed to be applied to humans in intransparent ways? An increasing number of people are extremely concerned about such developments and see considerable risks of such an approach. To address these concerns, city-led and human-centred AI approaches have recently been pursued.

### 2.3. City-Led Approaches: City Councils Taking Back Control

To some extent, smart cities intended to collect lots of data about everything and everyone, to optimise all processes (including human decisions and behaviours) with supercomputers, and to impose the supposedly "best solution" on the entire system utilising AI, have failed. Why is this, however? Such an approach typically works well when applied to production processes and logistics. However, while a business often pursues a particular goal, a city is not a business—and should not be run like one. It should rather enable and catalyse certain socio-economic processes.

The reason is simple: Determining a goal function that reflects all needs of society regarding prosperity, fairness, health, education, culture, environment, etc. is difficult, if not impossible. Different goals may even contradict each other. Therefore, cities try to reach different, often non-aligned goals in parallel, possibly in different locations.

Reducing this complexity into models forces urban management to operate on a limited understanding of cities. Therefore, these models need to be complemented by reason and experience [48]. This calls for the consideration of insights from the urban planning community. Overall, however, the complexity science and urban planning approach can be fruitfully combined with each other. Sennett [49], for example, interprets cities as open, non-linear systems, growing in unpredictable ways. Consequently, technologies should focus more on coordination than on command.

This requires space for political decisions and a city-led approach, which can be technologically supported. Here, rather than outsourcing city services to one company and letting it run the city like a company, the city decides about its goals, and each of them is addressed with one or more specifically tailored technological solutions.

### 2.4. Citizen-Centred Solutions

Until today, creating liveable cities is an art. Cities are not just giant optimisation problems, nor are they giant entertainment parks, where pre-manufactured experiences are consumed. Cities are places, where people meet, communicate, make friends, and fall in love. In other words, cities are first and foremost about people.

This has promoted the concept of trustworthy, explainable, human-centred AI [50,51]. Accordingly, AI solutions should serve humans, not the other way round. People—their needs and interests—should stand in the centre of AI applications. So far, the pursuit of this principle has mainly resulted in personalised information, products, and services.

However, these so-called citizen-centred solutions are often based on profiling, targeting and behavioural manipulation. In that way, they violate values such as privacy, informational self-determination and other fundamental rights [6,52], which are implied by the Universal Declaration of Human Rights [53].

The discussion about surveillance-based approaches in smart cities has been particularly fierce around methods of predictive policing. Great concerns were triggered not only by the high false positive rates of corresponding algorithms [54]. Moreover, the issues of systematic bias and discrimination have been raised [55,56], for example against people of colour [57] and other socially or economically disadvantaged people. Major discrimination was also found in face recognition algorithms [58]. As a result, California banned the use

of face recognition in public spaces—at least temporarily [59]. Furthermore, various big tech companies have decided to restrict the use of their face recognition software [60].

In summary, human-centred AI is not sufficient to satisfy the needs of citizens in smart cities, particularly collective (social and cultural) needs. Given the diversity of citizens, their individual goals and values may often oppose each other. However, amplifying contradictory goals and attempts to achieve them by technical means, as is often done by individually centred IT solutions, will undermine social cohesion. This may eventually lead to fragmented communities and broken societies, which finally may not be able to serve the goals and needs of humans well.

Hence, a human-centred AI approach as promoted today is insufficient. It lacks coordination capacity and the ability to promote collective intelligence. It also does not sufficiently support the emergence of shared values and collective action to address urban challenges that require cooperation and consensus. Therefore, future smart cities solutions should be focused on the needs of the citizenship and the civil society, not just human-centred in the sense of focusing on individuals.

### 2.5. Learning from Self-Controlled Traffic Lights

In recent years, some interesting lessons have been learned from the example of traffic light control in a city. This is an NP-hard optimisation problem, which cannot be exactly solved in real-time, even with supercomputers [61]. Hence, high performing solutions—typically synchronised periodic service patterns—are usually determined off-line, which are then applied in certain time windows or typical situations.

Such pre-determined solutions often suffer from large minute-to-minute, hour-to-hour, day-to-day, and week-to-week variability of traffic flows as well as from disruptions by accidents and building sites. Even adaptive traffic control schemes would not continuously change the order in which traffic lights are being served. They would typically shorten or extend the green times of a more or less synchronised, cyclical control scheme, which is changed over the course of the day [62]. Non-cyclical services are typically not considered.

Surprisingly, classical control strategies based on a traffic control centre and optimisation or machine learning approaches may be outperformed by a self-organised, decentralised traffic light control [63]. Such a self-control approach can be based on the flexible adaptation to local needs [64]. This is determined by short-term predictions considering the physics of queuing systems and traffic flows (hence, an "analytical approach") [65]. Overall, this approach enables extremely efficient management of limited resources such as spatial and flow capacities. Even though the exact timing of green phases is less predictable than for classical control approaches, average travel times are more predictable and shorter.

Altogether, the conclusion does not change, if machine learning approaches are being used [66]. These require a large computational effort to learn the complex dynamical features of a traffic flow network. Hence, when it comes to traffic light control we find:

1. Self-control approaches perform surprisingly well as compared to machine learning solutions.
2. In order to find superior solutions, one needs a "hybrid approach", where the scientific knowledge behind the analytical approach needs to be fed into the machine learning approach.
3. Even in the age of Artificial Intelligence, analytical approaches remain important, but hybrid approaches are best.

Altogether, in a complex dynamic system such as urban traffic flow, the flexible response to measurement-based short-term predictions of upcoming local demands can outperform classical planning and optimisation approaches, as the latter are often not adaptive enough to match real-life variability [64]. Similar kinds of challenges concern global and city logistics as well. Hence, the flexible management of scarce resources based on the response to short-term predictions of local needs is expected to lead to less wasteful, more sustainable solutions [67]. It is, furthermore, expected to contribute to the resilience of cities, i.e., the ability to recover from shocks and disruptions, disasters and crises. A

decentralised, adaptive approach is also more compatible with democratic principles than a centralised control approach imposed on the entire city.

## 3. Value-Sensitive Smart Cities

In the past years, it has become increasingly clear that technologies should be designed in compliance with constitutional and cultural values, because their use could otherwise damage the foundations of our society. This approach is called value-sensitive design. Batya Friedman and Peter Kahn [68–70] were the first ones to theorise the concept. Specifically tailored to the context of smart cities, Stone [71] proposes six principles to define and operationalise values. The following subsections reflect on his and further considerations.

In Section 3.1, we acknowledge that history matters, as the "war on terror" has brought along a trend of violating values. In Section 3.2 we choose Stone [71]'s approach of "valuableness"; we do not use abstract definitions of values, but rather highlight what is important to citizens. In Section 3.3 we give a concrete example that "specific technology matters". In Section 3.4, we stress that ethical issues are intertwined and, therefore, a plurality of values is needed to innovate responsibly ("boundary conditions"). Finally, Section 3.5 reflects on the fact that one must "abandon completeness": defining objective well-being, for instance, is difficult due to its inherent multi-faceted, sometimes competing dimensions.

### 3.1. Violated Values

Compared to the Universal Declaration of Human Rights [72], however, in many countries fundamental rights have been restricted ([73] provides an assessment of the measures taken concerning their impact on human rights across 11 jurisdictions). in the years 2020 and 2021, often with support of digital technologies. These restrictions include

- The right to life (Art. 3), given the application of triage;
- The rights of equality (Art. 1) and non-discrimination (Art. 7);
- The presumption of innocence until proven guilty in a public trial (Art. 11), given various predictive policing practices (see e.g., [74]);
- The protection of privacy (Art. 12), in view of mass surveillance;
- The freedom of movement (Art. 13), in view of geofencing applications and travel restrictions;
- The freedom of opinion and expression (Art. 19), in view of ongoing censorship on social media platforms;
- The freedom of peaceful assembly and association (Art. 20), in view of network shutdowns to interfere with assemblies [75].

While many of the restrictions have been attributed to the COVID-19 pandemics, there has been a trend in this direction at least since the "war on terror" [76] after 9 November 2001. Overall, the restrictions of freedoms and of human rights have affected human dignity altogether: in digital times, humans are increasingly being administered and managed like things, which should not be the case. The growing problem of hate speech, which often leads to real-world harm, also indicates that human dignity is affected. Furthermore, the circumstance that "code is law" [77] has undermined the principle that all political power should originate from The People (Art. 20, (2) GG). In a sense, "social engineers", who are not known to The People or accountable to them, have hacked society and now determine how it works.

In the digital age, fundamental principles such as *checks and balances*, which call for a division and decentralisation of power, have been compromised. While, in surveillance capitalism, some big IT companies have grabbed more and more power; in countries such as China, the ruling political party has taken over control. Overall, however, it seems that neither of these approaches have managed to create a favourable state of the world. The development has rather been paralleled by environmental degradation (often framed as the "climate emergency" [78]) and by a global loss of control in further essential matters such as international migration and health, or loss of species and diversity.

*3.2. Design for Values*

In view of this situation, experts have increasingly demanded a "value-sensitive design" (VSD) or "ethically aligned design" approach of digital platforms and technologies [79]. Accordingly, it is to be considered that, for a society to thrive, one needs to pay attention to a plurality of values [80,81]. It is important to note that efficiency and economic growth are not the only values that matter for societies. Environmental conditions and health, safety and security, human dignity, well-being and happiness, privacy and self-determination (autonomy, sovereignty, freedom), fairness, equality, and justice, consensus, peace, solidarity, sustainability, and resilience, for example (Figure 1), all need to be considered.

Friedman and Hendry [82] proposed seventeen VSD methods, among them "Stakeholder analysis" and "value source analysis". These two are particularly important to identify the spectrum of values that should be accounted for. The stakeholder criterion is a key concept in value-sensitive design, and it is also one of the four envisioning criteria [83]. The usage of envisioning cards by city-planners and policy makers can be a strong first step for design thinking for cities.

## DESIGN FOR VALUES

| | | |
|---|---|---|
| Privacy | Wellbeing | Solidarity |
| Autonomy | Safety | Peace |
| Equity | Security | Usability |
| Justice | Sustainability | Resilience |
| Dignity | Health | Efficiency |
| Happiness | Friendship | Flexibility |

**Figure 1.** When engaging in a "design for values" approach, it is important to consider that there is not only one value that matters. Different goals must be well balanced, which requires political negotiation and public deliberation. We want to stress that the list is neither mutually exclusive nor exhaustive.

Table 1 contrasts conventional smart cities with value-sensitive smart cities. In [84], for example, the authors investigate four smart city projects in Europe and find that collaboration, public–private engagement, and a bottom-up approach are at the heart of a successful smart cities. In another publication [85], the authors investigate how value-sensitive design offers hybrid approaches that can adapt to any existing space and can work in both directions (bottom-up and top-down). Smart cities of the future should promote a hybrid adaptive approach, where multiple stakeholders together establish priorities for policies that reflect the interests of a community. In this type of city, the use of technology is not limited to data collection, but creates a participatory ecosystem that promotes fairness, social innovation and democracy. The data ownership is with the people, who are the custodians of their data.

**Table 1.** A Conventional vs. Value-Sensitive Smart City.

|  | **Conventional Smart City** | **Value-Sensitive Smart City** |
|---|---|---|
| Design | Technology-centric | Value-sensitive |
| Organisation | Top-down control | Hybrid adaptive approach |
| Governance | Public–private partnerships | Multi-stakeholder, inclusive, and collaborative |
| Citizenship | Data producers | Active digital citizenship, informed consent |
| Technology | Data-driven | Ethical, citizen-centric |
| Data ownership | Proprietary data | Open data, digital rights |
| Dynamics | Political, corporate | Society-driven, social capital |

*3.3. Sustainable Innovation: An Example*

In the following, we illustrate the value-sensitive design approach for a couple of concrete examples. Let us start with the value of sustainability.

In 2019, the Dutch city of Utrecht introduced 300 new sustainable bus stops with plants- and grass-covered rooftops [86]. This generated considerable international interest. The green bus stops not only support biodiversity by attracting bees and other insects, but also capture fine dust particles, store rainwater and reduce heat stress in summer in the city. They are made of eco-friendly material, and their transparent design increases citizens' sense of security. What strikes us as smart and admirable about them is that a number of our values, namely sustainability, well-being, and security, are all realized in one coherent design.

There are two important ideas for the city in this specific example of sustainable innovation. First, there is the idea that our moral values can be designed for, that they can be embedded, incorporated, and exemplified in technology, in the same way as our values such as dignity, respect, equality, and justice can be expressed in the design of the constitution of a country or in its basic institutions. As recent work in value-sensitive design [82] has shown, moral ideas and ethical principles should not be seen as an obstacle. They can motivate or inspire designers, engineers and architects in their work to come up with new solutions that match requirements and specifications. Such innovations become "value sensitive" and go beyond optimisation. Reference [87] shows how this can generally work in architecture and the built environment.

The sustainable bus stop examples nicely illustrates that inter-, cross- and trans-disciplinary teams, for example, of applied scientists, engineers and designers, can learn to accommodate many values at the same time in one design. While each of them may feel morally overloaded by a multiplicity of values, together they often resist selecting a single value. Instead, they are committed to inventing new ways that allow them to satisfy as many values as they can. They aim for prosperity and clean energy. They want privacy and transparency. They are constantly pushing the limits of technology, because they want to design for all of the moral values at play. The individual is facing multiple values—often represented and advocated by different stake holders—and there often seems to be no obvious way to unify them, reduce them to each other, or resolve the conflict. Nevertheless, creativity and persistence may lead to superior solutions that were not imaginable before. Hence, responsible innovation can be seen as a moral challenge: it encompasses the introduction of novel functionality that allows one to satisfy more moral values than one could do without it.

*3.4. Preconditions and Design for Responsibility*

By explicitly, transparently and continuously designing for our plurality of values, we may hope to innovate responsibly. However, there is a "hidden" challenge in all of this that looms large: responsibility itself has become a design challenge. If we want to be able to take, hold, feel and make responsible decisions, we need to design for responsibility and accountability. This, in turn, requires design for knowledge, freedom of decision-making,

and control. In a world with ubiquitous AI, however, it cannot be assumed that the basic conditions for responsibility are satisfied. In fact, the use of deep learning approaches may seriously undermine responsible moral agency:

1.  Because of the black box character, AI may make it difficult for us to know what we are actually doing, resulting in a situation where it is always possible to say: "I did not really know what I was doing".
2.  Deep learning applications may also—as known from the literature on nudging and psychometrics—corrode our free choice and liberty. Thus, are choices really our choices any longer, or have individuals succumbed to AI-powered manipulation, big nudging, propaganda, and brainwashing? In that case, one could always truthfully say: "It was not a choice of my own. I was influenced, nudged or manipulated."
3.  A further condition for responsible agents, besides knowledge and freedom, is control. However, automated decision-making and autonomous systems are becoming more prevalent, not only in self driving cars. So the question becomes: "Are we in control?" Often the answer will be negative. This of course is a function of design. This issue has been addressed by Santoni de Sio and Van den Hoven [88]. The latter argue that meaningful human control over autonomous systems is a value sensitive design challenge, that has not yet been dealt with satisfactorily. It requires the articulation of an adequate notion of control and the specification thereof in terms of requirements for design of social–technical systems.

Therefore, we will have to come back later to the challenge of designing digital systems for freedom.

### 3.5. Measuring Happiness and Designing for Well-Being

We would also like to give examples how aspects such as happiness and well-being might be measured, given the availability of datasets about various aspects of society. For some time now, the gross domestic product (GDP), widely adopted as an indicator of well-being in society, has been criticised as misleading tool for public policy-making, since it is a poor proxy for a complex world. On one hand, defining objective well-being is difficult due to its inherent multi-faceted dimensions. However, the Organisation for Economic Co-operation and Development (OECD), the United Nations Development Programme (UNDP) and many national Statistics Bureaus have identified six major, observable ("objective") dimensions for the measurement of "good life": health, job opportunities, socioeconomic development, environment, safety, and politics. On the other hand, there is a "subjective" approach, which examines people's perception of their own lives, or happiness, defined by the degree to which an individual assesses the overall quality of their own life-as-a-whole favourably [89].

Traditionally, both, objective and subjective well-being, are measured with surveys of household income and consumption, an approach that carries strong scalability limitations. For example, surveys cannot provide dynamic accounts of well-being variations, cannot easily give a disaggregated picture of cities or neighbourhoods of metropolitan areas, cannot easily be stratified for groups, gender, age, or minorities, especially marginalised and vulnerable ones. That is why big data has been put forward as a complementary source, and scientific communities such as the European research infrastructure SoBigData (www.sobigdata.eu, accessed on 22 September 2021) have developed innovative methods to address the measurement of well-being and, more generally, to provide a Social Mining and Big Data Ecosystem for open, responsible Data Science [90,91]. Figure 2, from [91], illustrates the potential proxy connections between Big Data sources on the left and objective well-being dimensions on the right.

SoBigData and other communities focused on "Data Science for Social Good" have proposed many examples of sophisticated, privacy-protecting analytical processes such as

*   Nowcasting economic development on the scale of small areas, based on the diversity of human activities as measured by mobility data, such as anonymised GPS spatio–temporal trajectories and mobile phone call data records (CDR) [92,93];

- Nowcasting and forecasting the dynamics of influenza epidemics leveraging anonymised consumption profiles inferred from supermarket retail records [94];
- Assessing migrants' integration patterns using anonymised supermarket retail records [95] or migrants' home and destination attachments using anonymised Twitter data [96],
- Inferring citizens' mobility profiles from car navigation data in order to inform strategies to maximise pollution reduction [97] or identify best locations for public services (e.g., recharge stations for electric vehicles) [98];
- Analysing anonymised social media conversations and news to infer the level of happiness, from country- to city-level [99,100].

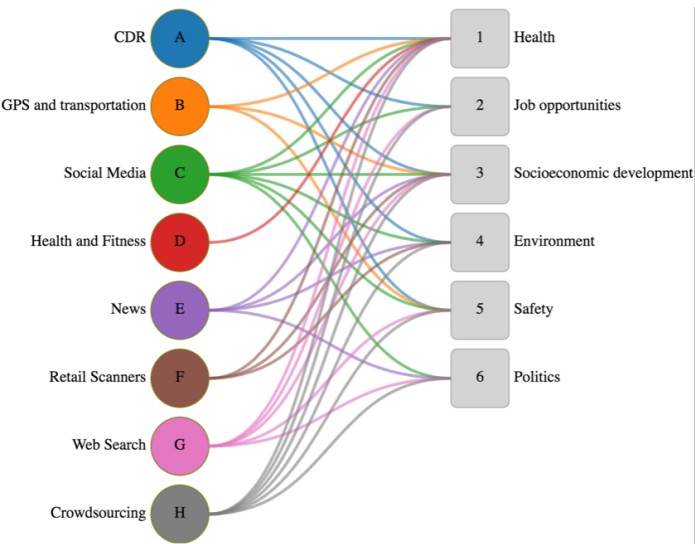

**Figure 2.** Bipartite network relating sources of Big Data (**left**) with the dimensions of objective well-being (**right**). Reproduced based on the Creative Commons Attribution 4.0 International License https://creativecommons.org/licenses/by/4.0/ (accessed on 26 August 2021) from [91].

The impact of big data analytics and social mining on cities has been widely discussed, e.g., in [101–103]. Of course, privacy-by-design approaches should be adopted in all of the above mentioned examples for social mining, to ensure that the methods used effectively protect personal information and individuals [104]. However, this is not enough.

## 4. Digital Sovereignty

From our point of view, personal data should ideally stay on personal devices. However, citizens should be enabled to easily collect detailed information about their location and movements, shopping transactions, social interactions and many other aspects of life. Furthermore, the users should be provided full control of such data together with the necessary tools to share only the information they want to share—at the preferred level of detail or aggregation. In fact, it should be easy to customise the sharing of information, depending on the individuals/entities with whom someone is interacting and depending on the purposes of data use [98].

Hence, our envisioned data management paradigm calls for a Personal Data Storage [105], where users are helped to collect and manage their data, equipped with data management and analytics tools for elaborating them, as well as with functionalities for controlling what kind of information—raw or derived from data—should be shared with other users or third parties, including public authorities. If properly designed, empowering citizens with these kinds of tools would enable active trustful participation in socio-economic, ecological, political and cultural affairs.

Accordingly, governments should ensure that

1.　Any data of personal relevance (also inferred metadata generated by transactions of any person with any kind of private or public institution) will also be shared with that person and included in the individual Personal Data Storage;
2.　Every person will be able to determine which company or institution is allowed to access, upon request, what part of the personal data, for what purposes and time;
3.　Every person will have a digital assistant that makes it easy to manage their own data (e.g., a personalised AI system on our smartphone that learns our data preferences);
4.　Systems will be designed with security in mind to prevent unauthorised access to personal data;
5.　Misuse of personal data will be punished (this includes the unauthorised use of personal data by third parties, which have not been intentionally made accessible to them for a particular use).

Then, the competition for data access would lead to a competition for trust and, thereby, to a trustable digital society. "Solid" seems to be one of the platforms trying to provide such functionality in the future [106].

Of course, one may still allow statistical evaluations of anonymised data for evidence-based governance [107–109]. However, procedural and algorithmic transparency will be needed [110].

### 4.1. Nervousnet Platform and Finance 4.0

The urgency of solutions for informational self-determination is amplified with the amount of data collected about people. To address the related concerns, it has been proposed to manage the Internet of Things in a distributed way. Figure 3 presents an overview of the proposed workflow for data management and privacy protection. Data collection should be done in privacy-preserving ways [111–113], e.g., data should be aggregated before they are stored or it should be deleted immediately after local real-time feedback was provided. Personalisation should happen on user devices, not on government or business servers. In this way, informational self-determination is enabled, as demanded in Section 4.

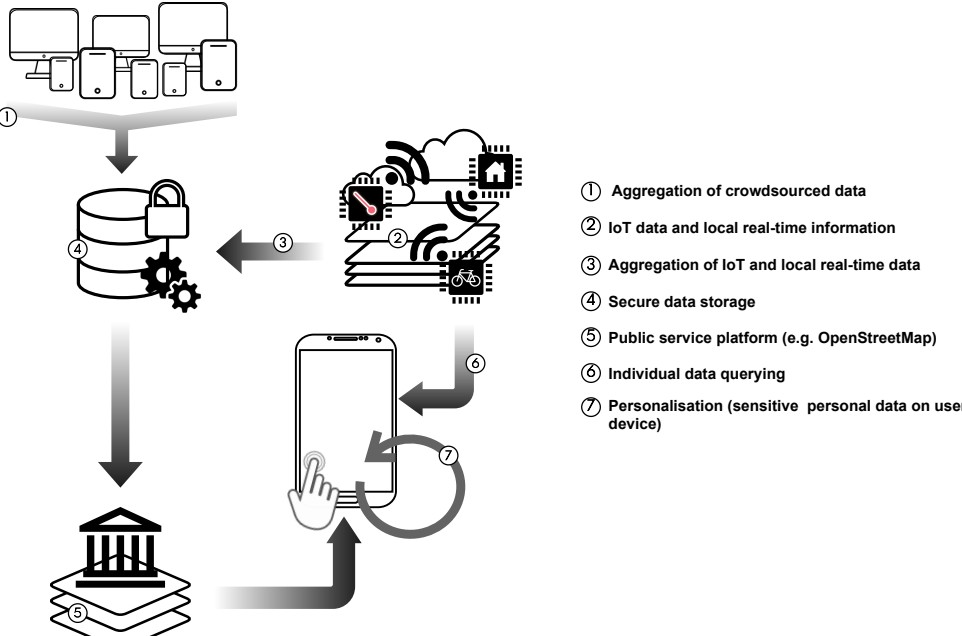

**Figure 3.** Proposed workflow for data management and privacy protection.

This participatory IoT approach has been the core of the "Nervousnet" project, which aimed to develop a public measurement network managed as a citizen web [7]. The col-

lectively generated data would be used to support greater awareness, coordination and adaptation [114].

The ideas for such a platform resulted from the preparations for the FuturICT flagship project [105]. Within the FuturICT 2.0 project, the Nervousnet concept was further extended by a multi-dimensional incentive system. This would consider externalities and risks, which are measurable by the Internet of Things or social proofs [115]. Specifically, the FuturICT 2.0 project proposed a socio-ecological finance system called "Finance 4.0" [116], for which a demonstrator (http://www.finfour.net, accessed on 28 August 2021) was built. The concept brings the Internet of Things together with Blockchain technology in order to create a multi-dimensional incentive system. It may be envisioned as a digital coordination system for complex adaptive systems that is based on various feedback effects.

Imagine, for example, that we would measure different kinds of externalities such as $CO_2$, temperature, or various resources utilising IoT sensors. In contrast to our current economic system, which valuates certain externalities by means of dollars, the Finance 4.0 system would do a separate accounting for different kinds of (measurement-based) currencies, which would not be easily convertible into each other (only for a considerable transaction fee/tax). This would establish a multi-dimensional finance system that would be better suited to incentivise environmentally friendly production and behaviour, friendly working conditions, or cultural engagement than our current monetary system. In a sense, it would add new forces to our socio-economic system, which would encourage, for example, the recycling and reuse of resources, thereby promoting the co-evolution towards a circular and sharing economy.

### 4.2. Human-in-the-Loop Approach

Figure 4 shows how one can create an IoT ecosystem, in which humans are central and have control over what data are being used and how. A collaborative ecosystem approach ensures that data do not serve exclusively corporate interests, but promotes collective intelligence and the interests of civil society. Priority is given to creating a network that connects to human values, and not just to "things" [117]. Such an ecosystem approach can bring the Internet of Things together with people in a way that creates a synergy between the city and the citizens.

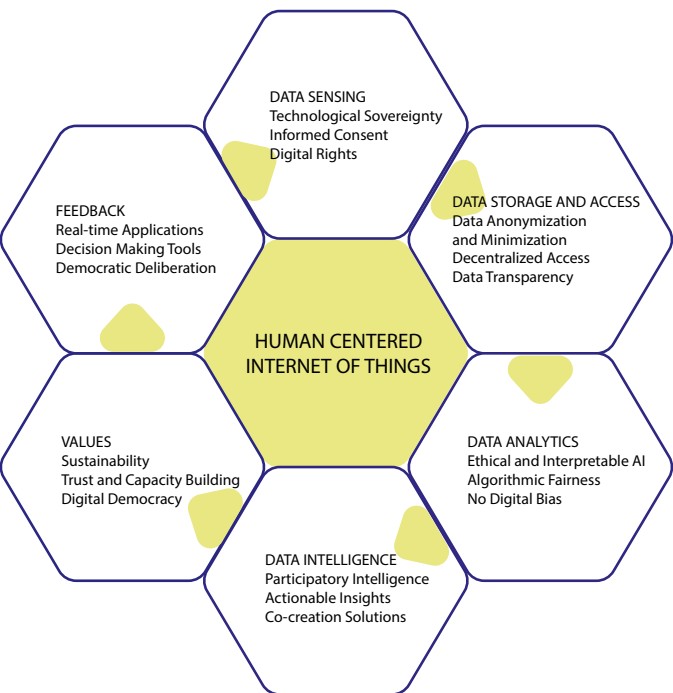

**Figure 4.** A citizen-centred IoT is at the core of a value-sensitive smart city.

There are several benefits of having a human-in-the-loop approach:

- A participatory design to understand the right problems to address;
- Aligning technology for long-term community use and sustainable development;
- Creating value for end users by providing better data services.

In some cases, however, it may not be enough to have a human in the loop. An example of this is triage apps [118], which were suggested to support medical doctors in taking life–death decisions Here, the problem is that recommendations of a data-based expert system may enact "epistemic authority" [119]. As a result, people may just execute what "intelligent machines" suggest, even though there might be serious scientific, legal, socio—political and ethical issues. In effect, this would mechanise death and eliminate the freedom of decision-making that human dignity is largely based on.

### 4.3. The Importance of Freedom

One of the essential preconditions for freedom is informational self-determination, as demanded above. As long as companies and secret services can determine people's ideas, decisions, and actions to a considerable extent, people are kind of remotely controlled and free decisions are largely an illusion. That is why we need solutions that comfortably allow for informational self-determination in the spirit of a *digital sovereignty approach*.

Freedom, in turn, is an important precondition for creativity and innovation, as innovation questions previous solutions. Nevertheless, it is increasingly common to let algorithms decide for humans, in favour of efficiency and process automation. As this can fundamentally undermine the freedom of decision-making, it is important for software engineers to carefully reflect what decisions can be automated without undesired side effects and what decisions a human being should take—or might want to take.

In fact, freedom of decision-making should be used particularly in "unclear" cases (where different indicators or goals suggest different decisions). Note that, even in situations of scarcity, freedom of decision-making does not need to be sacrificed. For instance, one may also imagine a system, in which all humans are granted a certain number of "vouchers", which they may use in the course of their life to override decisions that automated systems would otherwise take on them. The following subsection will provide a further example, how freedom can be exercised in situations, where resource constraints must be met.

### 4.4. Freedom vs. Optimality: The Example of Smart Grids

As pointed out before, digital technology is consuming a quickly increasing amount of energy [120], thereby undermining the sustainability goals [121]. Even though some well-known tech giants were among the first to commit to renewable energy use [122], most of them do not currently plan to cut back on their overall energy use. Recently, there are even growing concerns that peak demand from data centres may potentially drain resources needed to run services like schools and hospitals [123].

To overcome some of these energy-related issues, a complex networks approach can help to operate power grids as smart grids [124]. These provide a good example to discuss the issue of self-determination vs. optimality. From the perspective of energy generation, it is desirable (1) to cut peaks in the energy consumption and (2) to adjust to the fluctuating patterns of electricity generation (think of solar or wind energy). Therefore, political plans foresee the optimisation and control of the power consumption by citizens.

While some prefer a centralised control of energy production and consumption, others favour scalable, decentralised solutions, which flexibly respond to the local supply and demand. Those pushing for centralised control often argue with the "tragedy of the commons" [125]. They stress the failure of cooperation if everyone is selfish (i.e., non-cooperative), i.e., everyone shows the individually preferred consumption behaviour without considering others. According to them, an optimisation approach should determine a desirable consumption pattern, which would then be imposed on all consumers.

This, however, raises ethical concerns on various levels (for a discussion see e.g., [126]) and would contradict the principle of self-determination, and individual freedoms.

Figure 5 illustrates that centralised control of everyone's power consumption is not necessary to coordinate the latter. According to experiments, most people show some flexibility to adjust their energy consumption (the flexibility seems to be around 48% on average). The same figure also shows that fully cooperative behaviour is not required to avoid energy peaks. A semi-cooperative behaviour (50% flexibility for everyone) achieves the goal of cutting peaks almost as well as fully cooperative behaviour (requiring 100% flexibility). This speaks in favour of applying decentralised cooperation mechanisms rather than centralised optimisation, as it meets societal sustainability goals while providing the maximum amount of freedom and self-determination [127].

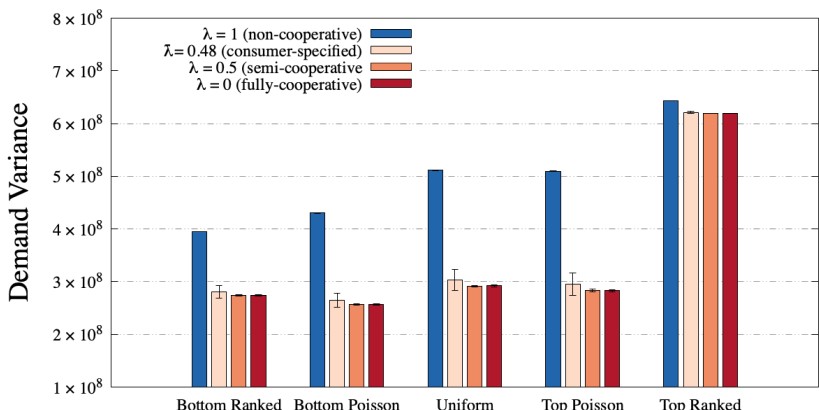

**Figure 5.** Variance in power demand depending on the cooperation strategy applied (adapted in compliance with the Attribution 4.0 International Creative Commons License, https://creativecommons.org/licenses/by/4.0/ (accessed on 22 August 2021) from [127]).

### 4.5. More Sustainable Consumption

Similarly, it is possible to support more sustainable consumption. Previously, it has been suggested to steer consumption patterns through personalised offers, nudging and neuro-marketing [128]. However, such approaches, based on behavioural control attempts (even when they are euphemistically framed as "liberal paternalism"), contradict democratic values and human rights [129,130].

Recent research has discovered alternative approaches promoting sustainable consumption. It turns out that consumers do not only care about the price of a product, but also about quality, health, environment, and social issues. Therefore, desirable behavioural change towards more sustainable consumption can also be supported based on self-determination, using value-sensitive digital shopping assistants [131]. Such assistants (see Figure 6) would consider the stated preferences of users, while evaluating specifics of products and ranking them, using data from various databases. This creates a three-fold benefit: (1) users find and consume products that match their preferences better. (2) Based on the aggregated user preferences, businesses learn to produce better products, which can be sold at a higher price. (3) Nature and health benefit as well.

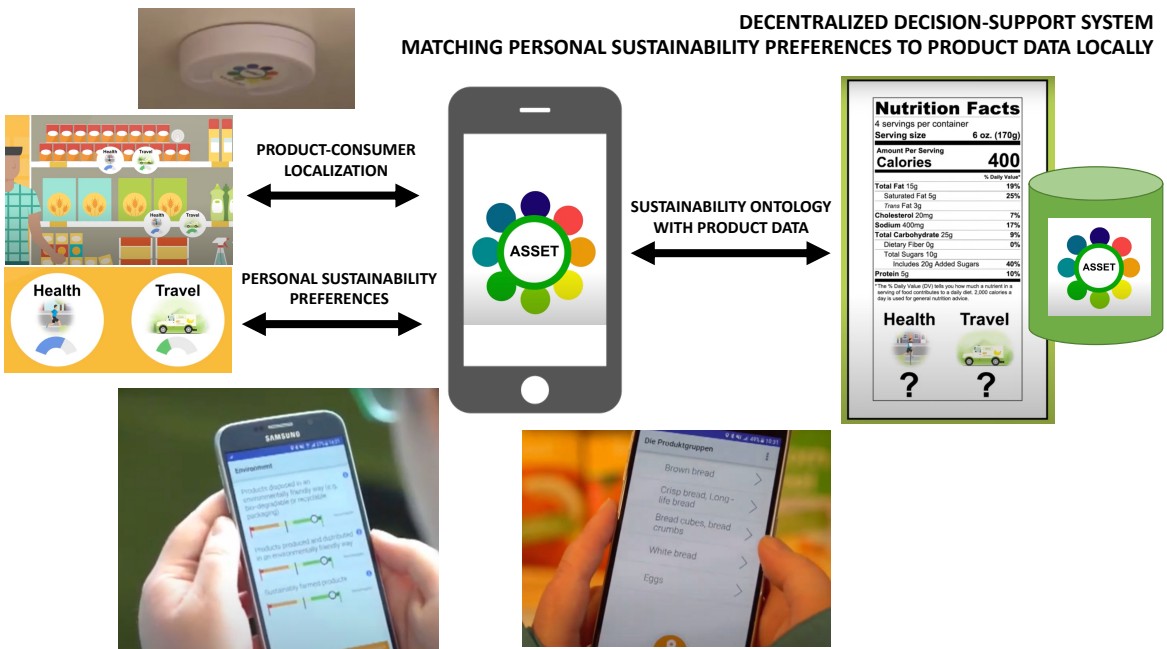

**Figure 6.** Illustration of elements of a value-sensitive personal shopping assistant to support more sustainable consumption, which was developed in the ASSET project. For an explanatory video watch https://www.youtube.com/watch?v=uur5 BXXspgI (accessed on 22 August 2021).

## 5. Democracy by Design

In the 21st century, our cities will need to deal with a long list of grand challenges. Design solutions are paramount. However, they will not add up to a liveable and just city unless we successfully design them for the values of well-being and justice. Reference [132] discusses the design for values in AI, particularly for human rights.

If we want governance and decision-making to be democratic, we need to have or to carve out and specify a conception of democracy that suits us and then shape information technology to support that.

In the following, we will reflect on "democracy by design". This approach aims to manage complex dynamical systems in a participatory way [107–109]—or even to build autonomously self-organising and self-regulating systems, based on complexity science and evolutionary game theory, mechanism design, and network engineering [52]. From the point of view of citizens, it would certainly be desirable if political leaders institutionalised efforts towards the creation of digital platforms that support civic participation, coordination, and self-organisation [133].

### 5.1. Civic Deliberation

A key element of "democracy by design" is to acknowledge the fact that every citizen should have the right to express their opinion on the systems that affect them to improve them. To allow for this, suitable methods and tools for civic deliberation and decision making need to be created. These should serve not only to promote meaningful interactions between multiple stakeholders but also to create trust between the citizens and those who govern them.

Civic deliberation can help to overcome conflicts and promote consensus. This discursive approach to ethics is considered to be critical for the moral development of the Smart City by other scholars, too [134].

Consensus does not have to be engineered by manipulating opinions, as this has been often done since the days of Edward Bernays [135] and his book on propaganda [136]. Given suitable interaction mechanisms, repeated interactions will naturally lead to the

emergence of self-organised conventions or social norms [137]. This applies to situations with different preferences as well (such as multi-cultural, conflict-prone settings) [138,139].

### 5.2. Harnessing Collective Intelligence

For practical implementation, platforms for Massive Open Online Deliberation (MOOCs) were proposed [114]. These aim to find innovative solutions that work for more people, obtained by a structured deliberation process. Inspired by the functional principles behind swarm intelligence [140], the process fostering collective intelligence involves different stages: (i) independent exploration, (ii) information exchange, and (iii) integration of solutions [52].

In social systems, an additional voting step (iv) to select one of several integrated solutions may follow, if no full convergence was achieved. However, the mechanism in the fourth step has to be chosen with care: Research shows that the voting rule itself directly influences the quality of the outcome [141]. Therefore, the voting rule should be designed in such a way that voters can express their true preferences [142]. In other words, rather than using a simple majority rule, one should consider multi-option preferential voting rules [143], which have a higher probability of selecting an option that works for more people (including minorities). Such rules can increase the social welfare [144], which is desirable.

We would like to point out that all four steps mentioned above may be supported by properly designed digital platforms [145]. Furthermore, suitable incentive systems may improve performance beyond traditional democratic and market-based solutions [146].

### 5.3. Benefits of a Participatory Approach

In the meantime, the great potential of Massive Open Online Deliberations (MOODs) [114] has been demonstrated in real-world settings. Taiwan's digital democracy has successfully applied it many times, using the POL.IS platform [147]. It matches people with different points of view and tasks them to find suitable compromises and solutions. When sufficient consensus is reached, the discussion enters the political stage and the process of law-making.

With this novel approach, Taiwan sets an example of how societies can be digitally upgraded in a way that promotes collective intelligence. By contrast, most current social media platforms today promote an "opinion war" rather than constructive dialogue, deliberation, and consensus. One possible side effect of this "attention economics" approach is the spread of fake news and hate speech. However, to achieve a "wisdom of crowds" [148–151] rather than a "madness of crowds" [152], it is important to give room for diverse opinions and avoid manipulation [153].

Well organised civic deliberation assures that the digital transformation does not undermine democracy, but rather strengthens it. Not only does it serve to raise the transparency and legitimacy of governance, but also to increase the participation and satisfaction of people. It further supports resilience, i.e., the ability of societies to flexibly adjust to surprises, challenges, and crises.

Resilience is promoted, for example, by diversity and decentralised organisation [154] as well as by digital assistance, fairness, and solidarity [155]. Particularly the recent concept of "participatory resilience" is based on bottom-up empowerment, enabling people to help themselves and support each other. Based on suitable tools (e.g., https://www.unocha.org, accessed on 17 August 2021), this concept is expected to increase the crisis-response capacity of societies considerably and even to lead to "anti-fragility" [156].

### 5.4. Collective Learning ("Co-Learning")

A human-centred smart city with built-in human values would foster learning not only on an individual level but also on a collective level. The EPOS project [115,157] is a research endeavour harnessing this concept. There, collective learning is realised within decision-support systems that assist citizens to choose among several options.

However, a choice recommendation is not just personalised to individuals. It considers several citizens. Moreover, decisions are incrementally updated as a result of exchanging aggregated data of previous choices, without revealing sensitive personal information or relying on a centralised authority. This is to avoid the risks of systematic, large-scale manipulation of decisions. The intended outcome of the system is emergent consensus as a result of human–machine symbiosis and collective intelligence that involves humans and intelligent machines.

Note that collective learning ("co-learning") has already been successfully applied to several smart city application scenarios [115]:

1.  *Transport systems:* by coordinating route choices of vehicles, traffic jams can be prevented and overcrowded, polluted city centres can be alleviated, while safety, fuel consumption, and driving comfort can be improved [158].
2.  *Energy consumption and production:* by coordinating the use of home appliances or the charging of electric vehicles, power systems can be made more reliable and decarbonised, shifting demand to off-peak times or to times with high power production from renewables [157].
3.  *Bike sharing:* the choice of bike sharing stations from which bikes are picked up or left can be coordinated to avoid bike sharing stations that are overloaded or left without bikes. These imbalances increase the operational costs of bike sharing infrastructures requiring manual reallocation of bikes at the end of the day [157].
4.  *Urban commons:* the shared use of libraries, shared apartments (Airbnb) or parking spaces can be coordinated to prevent the over-exploitation of public resources, e.g., overcrowded libraries, overcrowded city centres, parking spaces, etc. [115].

The EPOS algorithm is open-source (Available at https://github.com/epournaras/EPOS (last accessed: 10 May 2021)) and has been crash-tested for its robust operation of a high Technology Readiness Level (TRL ≥ 6) in decentralised and highly volatile environments [159].

### 5.5. Co-Creation and Open Innovation

Smart cities of the future should offer spaces that support deliberative processes, namely for the sake of fairer community representation, better informed decisions, and more meaningful outcomes. For example, "Living Labs" [160] and "Maker Spaces" [8,161] bring together multiple stakeholders and promote open innovation by creating a fair and inclusive environment, where technology and innovation come together in real-life contexts. This can create various benefits: (i) diverse people see problems from different perspectives, providing together a more complete picture of complex problems, (ii) local knowledge will be considered and local resources better used, (iii) the potential of neighbourhood communities will be mobilised, (iv) solutions can be developed for places that are not well reached by politics and business (this is of concern not only for marginalised communities).

Over the years, an increasing number of cities have been using such "open innovation" platforms to tackle challenging issues like air pollution [162], noise mapping [163] and radiation monitoring [164]. Some initiatives, such as "Make City" [165] or "Open Source Urbanism" [9,166–168] do not leave it there, but truly co-create the city. Importantly, the idea of co-creation can overcome classical hierarchies, and thereby promote respectful eye-level exchange between policy makers, researchers, citizens, and other stakeholders. This can support social innovation in the sense of responsible innovation in societally relevant contexts [169].

Note that such kinds of exchange are desirable if the equality principle shall be filled with life. It is also important to realise that equality is a principle that can promote optimality in self-organising systems [170]. In connection with the call for participatory opportunities, this brings us to the co-* principles [171]. These success principles for complex, networked societies include, for example, co-learning, co-creation, co-ordination, co-operation, and co-evolution (Figure 7).

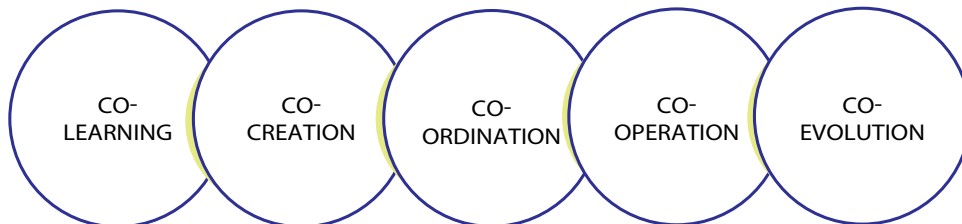

**Figure 7.** Illustration of some co-* principles.

In favour of higher legitimacy and societal progress, private businesses as well should learn to pursue their profit-oriented goals in ways that fit society's collective goals [172]. Such goals are very important for a thriving society, while they easily are undermined. This is why we need a suitable governance framework—one which takes externalities and systemic risks into account. [133] have coined the term "interactive governance" for joint efforts to address public issues that produce (more, smarter) public value. They define it formally as "the complex process through which a plurality of actors with diverging interests interact in order to formulate, promote and achieve common objectives by mobilising, exchanging and deploying a range of ideas, rules, and resources".

*5.6. City Challenges: A Scalable Approach to Address Global Challenges*

Finally, how can we address the challenges our world is faced with? Collaboration networks among cities are suited to unleash collective intelligence, allowing one to find solutions for global problems. Collaboration can further be catalysed through friendly competition. For example, ideas for Climate (and) City Olympics [173,174] have, in the meantime, been experimented with in various promising real-life settings [175–179].

When combined with principles of "open innovation", they could turn the regions of the world into giant maker spaces and innovation motors. This approach promotes a new kind of globalisation, called "glocalisation" [180], which is based on thinking global while acting local (and diverse). Such co-opetition frameworks (combining competition with cooperation) promise great benefits for the future.

**6. Summary, Conclusions, Discussion, and Outlook**

Traditionally, when people reflect on ethical issues, they do not think primarily of cities. However, cities are the places, where many problems of the world are concentrated—and solved. So, we should spend more time reflecting on what it means to design, build and operate cities ethically? In this connection, one immediately thinks of the problem of sustainability. However, there are a lot more ethical issues to consider. In this paper, we could only explore a few. Therefore, we hope that it can trigger a lively discussion about what is life about (in cities and beyond), and what we can do to improve the quality of life, proposedly in harmony with nature?

In response to this question, our paper has offered several puzzle pieces. We have argued for responsible innovation—in other words, to design for values—through a value-sensitive design approach [68–70].

We would like to point out that ethics should not be considered as an obstacle to desirable solutions. In many cases, design for values will deliver better solutions, which satisfy the needs and expectations of more people. Such solutions should not be "subtractive" in the sense of "bad compromises" (the least common denominator). They should be "good compromises", as they result from collective intelligence. For collective intelligence to emerge, however, certain organisational principles must be applied, as was mentioned above.

In recent years, we have seen that politics started to appreciate "citizen councils" as a valuable framework to solve difficult societal problems, where classical politics faces limitations. We have also seen the emergence of platforms such as POL.is [147], Consul [181], LiquidDemocracy [182], or SmartCitizen.me [183], which are encouraging the constructive

engagement of citizens. These enable more participatory forms of democracy, powered by digital technology, in other words: digital democracy. Such approaches are also relevant for participatory sustainability [116,131] and participatory resilience [155].

As we have pointed out, solutions designed for values do not need to be inefficient. They just work differently. At the beginning of this paper, we have stressed that planning, optimisation and control may not be the best approaches to manage complex dynamical systems, which are often characterised by feedback, side and cascading effects as well as large variability, randomness, uncertainty, and disruptions. Such systems are frequent in our hyperconnected, globalised world, and managed better based on a flexible response to local needs. In fact, such solution approaches can be a lot more resilient, while being efficient and compatible with fundamental democratic principles.

We point out that self-organisation, self-regulation, and self-control are widespread principles in biology. It is, therefore, interesting to note that nature has already managed to establish a "circular economy" despite the absence of centralised control. We might learn a lot from organisational principles of biological and ecological systems. It is for such reasons that researchers have suggested organising the digital world in a participatory way, as an "information ecosystem" [184]. If designed well, such a system would be beneficial for all, by fostering synergy effects.

When addressing the ethical issues of smart cities, we should turn away from the exploitation of people and nature that is common today and engage in a new paradigm based on mutually beneficial relations [185]. Again, symbiotic interactions in nature may serve as a source of inspiration. In view of the problems of the world, a novel approach is needed—a new paradigm that moves away from the manipulation and control of people towards citizen empowerment and coordination. If properly done, this will be able to mobilise civil society for change for the better. Our vision is one of "synergistic intelligence", which combines artificial and human intelligence in beneficial ways and leads to cooperative benefits. Now, digital technologies can help to catalyse this favourable kind of organisation, if used well and designed for suitable values.

**Author Contributions:** Conceptualisation, D.H.; funding acquisition, D.H., F.G., R.H., J.v.d.H., D.P. and E.P.; investigation, D.H., F.F., F.G., C.I.H., J.v.d.H., S.M., D.P. and E.P.; project administration, D.H., C.I.H., S.M. and E.P.; supervision, D.H.; visualisation, F.F., C.I.H. and S.M.; writing—original draft, D.H., C.I.H. and S.M.; writing—review and editing, D.H., R.H., C.I.H., S.M. and E.P. All authors have read and agreed to the published version of the manuscript.

**Funding:** D.H., R.H. and E.P. acknowledge support by the SNF project on "Decision-making process supported by a participatory platform: Consequences on trust, on legitimacy of the political decision, and user skills" through the NRP77 "Digital Transformation" (project no. 407740_187249). F.G. and D.P. acknowledge support through the project "SOBIGDATA++: European Integrated Infrastructure for Social Mining and Big Data Analytics", which has received funding from the European Union's Horizon 2020 research and innovation programme under grant agreement No. 871042. J.v.d.H. acknowledges support through the project "HumanE AI Network", which has received funding from the European Union's Horizon 2020 research and innovation programme under grant agreement No. 952026. C.I.H. and S.M. acknowledge support through the project "CoCi: Co-Evolving City Life", which has received funding from the European Research Council (ERC) under the European Union's Horizon 2020 research and innovation programme under grant agreement No. 833168.

**Institutional Review Board Statement:** Not applicable.

**Informed Consent Statement:** Not applicable.

**Data Availability Statement:** Not applicable.

**Acknowledgments:** We would like to thank the following people: Figure 2 was reproduced from a paper by Vasiliki Voukelatou, Lorenzo Gabrielli, Ioanna Miliou, Stefano Cresci, Rajesh Sharma, Maurizio Tesconi and Luca Pappalardo based on the Creative Commons Attribution 4.0 International License. Thomas Asikis and Johannes Klinglmayr were involved in the ASSET project for more sustainable consumption. Marcin Korecki worked on combining the analytical self-control approach for traffic lights with machine learning approaches. Javier Argota Sánchez-Vaquerizo contributed

to the data management flowchart in Section 3 and gave paper recommendations on the early use of computers for cities. The authors would also like to thank all researchers and developers who have contributed code to the Nervousnet repositories (https://github.com/nervousnet, accessed on 15 August 2021) and to the Finance 4.0 project (see [116] for details) as well as to the EPOS project [145,157,159].

**Conflicts of Interest:** The authors declare no conflict of interest. The funders had no role in the design of the study; in the collection, analyses, or interpretation of data; in the writing of the manuscript, or in the decision to publish the results.

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
