# Peer review of "Ethics of Smart Cities: Towards Value-Sensitive Design and Co-Evolving City Life"

_sustainability, doi:10.3390/su132011162_

Round 1

Reviewer 1 Report

This paper attempts to explore the VSD approach to smart city design and co-creation.

To begin, the paper is not immediately clear as to what it aims to discuss. The abstract seems to simply make a series of statements without any proposition, arguments or substantive conclusions. The authors need to more clearly state the ‘problem’, what their paper aims to do and what conclusions they draw.

Similarly,  There is no clear structure to this paper at all. The authors simply make a series of statements with no clear direction, or any direction at all. The paper is missing a clear introduction that lays out what the abstract aims to summarize, that is: (1) the problem/background, (2) what the main arguments that the paper aims to propose and argue for, (3) the aims of the paper. Likewise, in privation of this, it seems that the reader is also unable to determine the comparative originality of this paper and thus its contribution to the literature (what literature does this even belong to?). Similarly, the authors should be very clear as to the literature gap that this paper aims to fill. It is not until page 6 where the authors state, at least more or less clearly, what their paper aims to do. This should be present both in the abstract and at the beginning of the paper.

With regards to VSD itself, the authors never give any clear guidance as to exactly what VSD approach is needed. It has been over two decades since VSD has been considered a monolithic methodology. It has, since, evolved into an umbrella term that encompasses more than 17 different methodologies, each having its own unique boons and constraints. Simply throwing the term VSD as an approach is not sufficient, the authors need to argue for which and why such approaches are suitable to the examples chosen rather than simply using VSD wholesale. For example, envisioning cards may be a strong first step for long-term, multi-generational design thinking for sociotechnical systems (i.e., the whole of smart cities) (c.f., Friedman et al., 2012).

This issue, despite being persistent throughout the paper, does not sideline some of the important and timely issues that the authors bring up. That being said, the paper offers some useful and novel insights into how VSD can be applied more broadly to the design of smart cities, however, baring major revisions this paper cannot be published as is given some major blindsides to the way it has been written and the privation of some foundational literature that cannot be ignored.

Specifics:

Page 6: the chart you use utilizes values that are mostly referenced in the VSD as pertaining to HCI rather than all technologies in particular. Although most, if not all, of them pertain to smart cities, it should be noted that the list is not mutually exclusive nor exhaustive of other values.

Page 6: in Table 1 you say that VSD organization is bottom-up. However, this is not necessarily so. What evidence do you provide to justify this table? VSD offers hybridized approaches to design that allow for both of those strategies (top-down and bottom-up) to take place. One of the benefits of VSD is that it adapts itself to any existent design space, even top-down ones. See for example Umbrello and Gambelin (2021)

Page 7: in section 2.3 you discuss some of the issues emerging for values from AI. However, you should take into account existing literature on VSD and AI which draws on many of the issues you raise. For example, Umbrello and van de Poel (2021) discuss the unique challenges that AI poses for VSD. This is something you should engage with.

Likewise, in #3 of section 2.3 you scuss the notion of control and says that “often the answer will be negative”. This of course is a function of design. You can see that this issue has been addressed already at length by Santoni de Sio and van den Hoven (2018) and Umbrello (2021).

References

Friedman, B., & Hendry, D. (2012, May). The envisioning cards: a toolkit for catalyzing humanistic and technical imaginations. In Proceedings of the SIGCHI conference on human factors in computing systems (pp. 1145-1148).

Santoni de Sio, F., & Van den Hoven, J. (2018). Meaningful human control over autonomous systems: A philosophical account. Frontiers in Robotics and AI5, 15.

Umbrello, S. (2021). Coupling levels of abstraction in understanding meaningful human control of autonomous weapons: A two-tiered approach. Ethics and Information Technology, 1-10.

Umbrello, S., & Gambelin, O. Agile as a Vehicle for Values: A Value Sensitive Design Toolkit.

Umbrello, S., van de Poel, I. Mapping value sensitive design onto AI for social good principles. AI Ethics 1, 283–296 (2021). https://doi.org/10.1007/s43681-021-00038-3

Reviewer 2 Report

Overall, the article reflects upon and develops an important idea: how to move beyond current/existing ideologies and discourses on the ‘smart’ city and re-think how to utilize high-tech innovations such as IoT and AI through a ‘value-sensitive’ city. It does so via a wide-ranging discussion of relevant issues, supported by a variety of interesting and relevant resources. The paper opens many questions and topics without providing specific or detailed directions – however, read as a position paper and vision for re-orienting the smart city, it is understandable to leave the details for future papers and debates. As such, I think this paper is an important contribution to VSD/smart city literature and should be published.

That said, I have a few comments and suggestions for the authors, which I (as always) is intended as constructive criticism to assist in improving the final version:

First, there are many minor typos and/or grammatical errors throughout the paper. Before publishing, the authors should undertake a careful read to clean up the text.

Second, the paper covers quite a bit of topics and jumps right in – as a reader, I was admittedly a bit lost in Section 1 regarding the overall thesis (which, I believe, comes at the end of 1.4?). The paper would benefit from a (brief) introduction preceding what is currently Section 1, to more clearly present the thesis and structure/narrative of the paper. This would serve to orient the reader, and likely also crystalize the (various) contributions made about the potential of a value-sensitive smart city.

Third, and related to the above point about a stronger introduction/narrative: there is some connection to smart city and city science literature, but little discussion of aligned ideas from urban planning. For example, Richard Sennett’s notion of the coordinating smart city or Sarah Williams work on using data for social good. Some (brief) mention of how a value-sensitive approach expands, advances, supports, contests, or otherwise these approaches would help to further clarify this paper’s contributions.

Round 2

Reviewer 1 Report

After reviewing the authors' revisions in light with the reviewers' comments, the authors have made sufficient changes and I see no further reasons to bar publication.